# Hepatitis A Immunity and Paediatric Liver Transplantation—A Single-Centre Analysis

**DOI:** 10.3390/children9121953

**Published:** 2022-12-12

**Authors:** Tobias Laue, Johanna Ohlendorf, Christoph Leiskau, Ulrich Baumann

**Affiliations:** 1Division of Paediatric Gastroenterology and Hepatology, Department of Paediatric Liver, Kidney and Metabolic Diseases, Hannover Medical School, 30625 Hannover, Germany; 2Paediatric Gastroenterology, Department of Paediatrics and Adolescent Medicine, University Medical Centre Göttingen, Georg August University Göttingen, 37073 Göttingen, Germany

**Keywords:** paediatric liver transplantation, chronic liver disease, vaccination, immunisation, hepatitis A, HAV

## Abstract

Following paediatric solid organ liver transplantation, risk of infection is high, both in the short and long term. Even though an infection with hepatitis A virus (HAV) is often asymptomatic and self-limited in children, some case studies describe severe cases leading to death. Vaccinations offer simple, safe and cheap protection. However, data on vaccination rates against hepatitis A in children with liver disease are scarce. Moreover, the vaccine is only approved from the age of one year old. At the same time, up to 30% of children with liver disease are transplanted within the first year of life, so the window of opportunity for vaccination is limited. This retrospective, observational, single-centre study examines the HAV immunity in paediatric liver transplant recipients before and after the first year of transplantation. Vaccination records of 229 of 279 (82.1%) children transplanted between January 2003 and June 2021 were analysed. Of 139 eligible children aged ≥ 1 year old, only 58 (41.7%) were vaccinated at least with one HAV dose prior to transplantation. In addition, seven patients received the vaccine below one year of age. After one or two doses, 38.5% or 90.6% of 65 patients were anti-HAV-IgG positive, respectively. This percentage remained stable up to the first annual check-up. For children vaccinated only once, a shorter interval from vaccination to transplantation is a risk factor for lack of immunity. Thus, HAV immunisation should be started earlier in liver transplant candidates to improve immunity in this high-risk group.

## 1. Introduction

An infection with hepatitis A virus (HAV) accounts for half of viral hepatitis cases in the United States [1]. Within the European Union/European Economic Area, the incidence is 2.5 cases per 100,000 population, but this is subject to regional variations [2]. Due to the often asymptomatic course of infection in patients under 6 years of age, the incidence may be underestimated [1,2]. With increasing age, the likelihood of symptoms rises with fever, abdominal pain, nausea and vomiting but in most cases the infection is self-limiting [1]. However, case reports describe severe courses resulting in acute liver failure with hepatic encephalopathy and the need for liver transplantation or leading to death [3].

The hepatitis A vaccine was introduced in 1992 [4]. Long-term data show high immunogenicity after basic immunisation in healthy children even after 10 years [5,6] and in >97% of adults after 20 years [7]. As infants usually have a mild or even asymptomatic HAV infection, vaccination is usually recommended from 12 months of age [4]. The introduction of hepatitis A vaccination to the regular vaccination programme in the United States led to an over 90% incidence reduction [8,9]. Moreover, it led to a decrease in hepatitis A-related fulminant liver failure, which resulted in reduced liver transplants in children [10]. However, recommendations for primary hepatitis A vaccination vary considerably between countries, especially due to the fact that in infants the disease is often asymptomatic or mild [4]. In Germany, it is recommended that children with chronic liver disease receive hepatitis A vaccination, but this is not part of the national childhood vaccination program. As a result, children with chronic liver disease need to be followed up with regard to their indication vaccinations: in a European survey, only 40% of children [11] and in the US, 68% of paediatric liver transplant recipients are vaccinated against hepatitis A at the time of liver transplantation [12]. However, immunogenicity appears to be lower in children with liver disease compared to healthy individuals [13]. In addition, there appears to be a decrease in antibodies against HAV, as only 39% of those vaccinated once before transplantation still have detectable antibodies one year after transplantation [14]. Nevertheless, there is a lack of data on long-term immunity in this high-risk population under immunosuppression.

This retrospective, observational, single-centre study investigates the hepatitis A immunisation status in paediatric liver transplant recipients. The objective of the study is to determine the immunogenicity of pre-liver-transplant HAV vaccination in children, both before and after the first year of transplantation, and influential factors for non-immunity.

## 2. Materials and Methods

### 2.1. Patients and Data Acquisition

In this single-centre, retrospective, observational study, data on children who received liver transplantation at Hannover Medical School (Germany) between January 2003 and June 2021 was analysed. Only patients who were under 16 years of age at the time of transplantation had a certified vaccination record and had received at least one hepatitis A immunisation prior to liver transplantation were included. Exclusion criteria were history of a post-transplant lymphoproliferative disorder (PTLD) and/or administration of B cell depletion therapy (e.g., rituximab), a history of infection with hepatitis A or the need for haematopoietic stem cell transplantation. In addition, anti-HAV-IgG were excluded from the analysis if the patient had received fresh frozen plasma or immunoglobulins within 3 months prior to measurement. Standard immunosuppression includes monotherapy with ciclosporine A (trough level from months 4 to 12 after transplantation: 120–160 µg/L) or tacrolimus (trough level from months 4 to 12 after transplantation: 4–8 µg/L). In patients with intensified immunosuppression, the calcineurin inhibitor is supplemented with mycophenolate mofetil (MMF) or prednisolone. The study was conducted according to the guidelines of the Declaration of Helsinki and approved by Ethics Committee of Hannover Medical School (Statement N° 9928_BO_K_2021, approval date 6 August 2021).

### 2.2. Chemiluminescence Microparticle Immunoassay for the Determination of Hepatitis A Immunity

Hepatitis A immunity was determined before and one year after transplantation at the first annual follow-up by analysing anti-HAV-IgG (ARCHITECT HAVAb-IgG) using the automated chemiluminescent microparticle immunoassays (CMIA) on the ARCHITECT platform (Abbott Laboratories, Abbott Park, Chicago, IL, USA). Briefly, antigen (HAV)-coated paramagnetic microparticles were mixed with serum samples from patients. An acridinium-labelled anti-human-IgG antibody conjugate was then added to generate chemiluminescent signals. Detection of anti-HAV-IgG in samples was performed by comparing the chemiluminescence signal with the cut-off signal of an active calibration. According to the manufacturer’s instructions, a sample to cut-off ratio (S/CO) ≥ 1.0 indicates a positive result, i.e., a past vaccination. The results are presented as qualitatively positive or negative.

### 2.3. Statistical Analysis

Qualitative data are expressed as numbers and percentages. Quantitative data are expressed as medians (25–75% quartile). The comparison of two groups with categorical variables was performed using chi-squared test, if the minimum requirements for expected frequency were met. Otherwise, Fisher’s exact test was used instead. Mann–Whitney U test was performed for continuous variables due to non-normality. Data was significant with *p*-values of *p* < 0.05 and marked as follows: * *p* < 0.05, ** *p* < 0.01, *** *p* < 0.001. Data was managed using R version 4.0.5 [15]. For graphical data, ggplot2 package version 3.3.3 was used [16].

## 3. Results

### 3.1. Study Population

Between January 2003 and June 2021, a total of 279 children received liver transplants in Hannover. Vaccination records of 229 (82.1%) patients were available. Since the hepatitis A vaccine is officially licensed in Germany from 1 year of age, vaccination would have been possible in 139 of the 229 children (60.7%). A total of 58 of 139 (41.7%) eligible children were vaccinated with at least one dose against hepatitis A before liver transplantation. In seven children, vaccination was started in the first year of life, resulting in 65 patients analysed of whom 41 (64.6%) were female. Median age at transplantation was 4.6 years (2.0–7.8).

A total of 46.2% of patients were diagnosed with biliary atresia (BA). Further diagnoses in order of frequency were progressive familial intrahepatic cholestasis (PFIC) (20%), cystic fibrosis (7.7%), cryptogenic cirrhosis (7.7%), Alagille syndrome (6.2%), metabolic conditions (4.6%), hepatic malignancy (1.5%), acute liver failure (1.5%), primary sclerosing cholangitis (1.5%), KIF12 variant, identified as a cholestasis-associated candidate gene, (1.5%) [17], and alpha-1 antitrypsin deficiency (1.5%). Baseline characteristics are presented in Table 1.

### 3.2. Hepatitis A Seroconversion Depending on Vaccination Doses Received before Liver Transplantation

Of these 65 children, 30 (46.1%) were vaccinated with one dose against hepatitis A before liver transplantation. Anti-HAV-IgG was measured in 26 of these children at the time of transplantation, of which 10 (38.5%) were positive. Of those children who had received basic immunisation with 2 doses, 32 out of 35 had an anti-HAV-IgG measured at the time of transplantation, of which 29 (90.6%) were positive, significantly more compared to one vaccination dose (Figure 1, *p* = 0.000026).

At the first year of follow-up after liver transplantation, 11 of 30 (36.7%) children still were anti-HAV-IgG positive after one dose, and 34 of 35 (97.1%) patients after 2 doses. Thus, there is no significant difference compared to the pre-transplant samples (*p* = 0.889951 and *p* = 0.260701, respectively). However, the percentage of anti-HAV-positive patients after one dose is significantly lower compared to those with complete basic immunisation with two doses (*p* < 0.00001).

### 3.3. Intraindividual Change in Hepatitis A Immunity from Prior to Post Liver Transplantation

In 58 of the 65 (89.2%) children examined, anti-HAV-IgG was measured both before and after transplantation. Of these, 39 (67.2%) patients were positive at the time of transplantation. Only in 3 of the 39 (7.7%) children, no anti-HAV-IgG could be detected at the first annual follow-up (Figure 2).

Next, factors influencing non-immunity after transplantation were investigated. Due to the small sample size, only Fisher’s exact test could be applied: only the number of vaccinations carried out prior to transplantation showed a significant difference, as all non-immune children had only received one HAV-dose before liver transplantation (Table 2, *p* < 0.0131).

### 3.4. Factors Influencing Hepatitis A Immunity in Liver Transplanted Children

In the next step, factors influencing hepatitis A immunity at the time of the first annual check-up were investigated. For this purpose, the patients were grouped according to their anti-HAV-IgG positivity into immune (*n* = 45) and non-immune (*n* = 20). As shown in Figure 1, there is a significant difference in the number of vaccine doses administered prior to transplantation (*p* < 0.00001). Furthermore, those children with serological immunity were significantly older at the time of transplantation (median age 5.54 [IQR 3.13–8.19] vs. 1.84 [0.81–5.05] years, *p* = 0.0008) and had a longer interval from last vaccination to transplantation (median 1.41 [0.56–2.65] vs. 0.18 [0.07–0.41] years, *p* < 0.00001). Children who are not immune received vaccination more often below 12 months of age (*p* = 0.000855). There were no differences in gender and trough levels of immunosuppression (including an intensified immunosuppression). Moreover, history of biopsy-proven acute rejection did not influence HAV immunity. The results are summarised in Table 3.

### 3.5. Factors Influencing Immunity after a Single Hepatitis A Dose

As only around one third of children are immune after one HAV-dose, we next investigated influential factors within our group. For this purpose, children were classified into immune (*n* = 11) and non-immune (*n* = 19) on the basis of anti-HAV-IgG positivity at the first annual check-up. The two groups did not differ significantly in age at vaccination and age at liver transplantation (Figure 3A,B). However, the interval from vaccination to liver transplantation is significantly shorter in the non-immune group compared to the immune group, as shown in Figure 3C (*p* = 0.00008). A significant influence of immunosuppression, history of biopsy-proven acute rejection and gender on non-immunity could not be demonstrated. Moreover, both groups did not differ in the number of children vaccinated under 12 months of age.

## 4. Discussion

This retrospective, observational, single-centre study investigates hepatitis A vaccination and serostatus at the time of and one year after paediatric liver transplantation. As hepatitis A vaccination is not part of the national childhood vaccination programme in Germany, only two of five children had been vaccinated at least once at the time of transplantation. This is in line with data from Switzerland, with over 40% of vaccinated children [14], but significantly less than in a study from the USA, where about two-thirds of all patients were vaccinated according to age [12]. This underlines the need to improve vaccination rates in this high-risk group.

Healthy children show seroconversion rates of 95% after at least two weeks following their first vaccine dose [4]. In a study by Ferreira et al., only 76% of paediatric patients with chronic liver disease showed seroconversion one month after their first immunisation compared to 94% of the healthy control group [13]. After the second dose, the seroconversion rates were similar with 97% and 100%, respectively. Our patients showed a decreased response with 38.5% and 90.6% anti-HAV-IgG positivity at transplantation after one and two doses, respectively. However, the interval between vaccination and measurement varied in our analysis and the patients in the cited study were, on average, older. A Swiss study showed HAV antibodies in 67% of children at transplantation, although it is unknown what the average number of vaccination doses was beforehand [18]. Nevertheless, this highlights the significantly poorer immune response in paediatric patients with liver disease to hepatitis A vaccination and underlines the need to complete the basic immunisation process to achieve a better outcome.

It should be emphasised that one year after transplantation and thus after the peak of immunosuppression, no significant decrease in antibody detection in both groups was observed: more than 36% and over 97% of children were anti-HAV-IgG positive at their first annual check-up after one and two vaccine doses, respectively. This is in line with a study by Diana et al., in which almost 39% of children still had antibodies after a single vaccination dose [14]. Furthermore, biopsy-proven acute rejections and (intensified) immunosuppression did not have any significant effects in our study.

Nevertheless, significant influential factors for the detection of anti-HAV-IgG after liver transplantation are the number of vaccination doses and a later age at liver transplantation. However, the latter seems to reflect the fact that older children have more time to receive vaccinations before transplantation. Looking at the data of the group of children who were vaccinated once before transplantation, only the interval from vaccination to transplantation is significantly shorter (Figure 3C, *p* = 0.00008). Vaccination in the first year of life, as in a total of 7 children, may be a risk factor, especially due to maternal anti-HAV-IgG levels which may reduce the vaccine immunogenicity [4], but did not show significant differences in our study in children vaccinated once prior to transplantation. Data on the effectiveness of vaccination in children younger than 12 months is scarce [19], probably due to the often mild disease course at that age. The Infectious Diseases Society of America (IDSA) guideline on the vaccination of immunocompromised children from 2013 recommends hepatitis A vaccination from 12 months of age [20]. The same applies to the CDC Immunisation Schedule, although vaccination can be administered earlier for certain high-risk groups. However, hepatitis A vaccination for infants between 6–11 months is only recommended for travel outside the United States [21]. Moreover, vaccination recommendations differ among paediatric hepatologists [22]. As up to 30% of children with liver disease receive liver transplants in their first year of life [23,24], earlier immunisation would, by contrast, offer the possibility of achieving a certain degree of protection. In addition, measuring anti-HAV-IgG levels in both mother and child would allow for a better assessment of whether vaccination is necessary.

Alongside the fact that this is a retrospective study, there are several other limitations: the chemiluminescence microparticle immunoassay used only detects anti-HAV-IgG, but no titres. Therefore, our study could not take into account antibody levels, which are lower in children with liver disease compared to healthy children after hepatitis A vaccination [13]. Moreover, the measurement of anti-HAV-IgG positivity was done at the time of transplantation and the interval between vaccination and measurement is therefore variable. Additionally, poor nutritional status, which is common in paediatric chronic cholestatic diseases [25], may possibly reduce vaccine efficacy [26]. In addition, several different hepatitis A vaccines are licensed in Germany, but with comparable seroconversion rates in healthy adults [27,28].

In summary, hepatitis A vaccination rates in children with liver disease are poor, despite the high risk of infection. Including HAV vaccination in the national childhood vaccination programme could improve these rates and reduce the incidence in the population [8,9]. Regular vaccination card checks with recommendations for immunisation must be carried out by paediatricians and hepatologists. Bringing forward the hepatitis A vaccination to include the first year of life may lengthen the time between vaccination and transplantation and increase seroconversion, consequently improving vaccination coverage in this high-risk group.

## Figures and Tables

**Figure 1 children-09-01953-f001:**
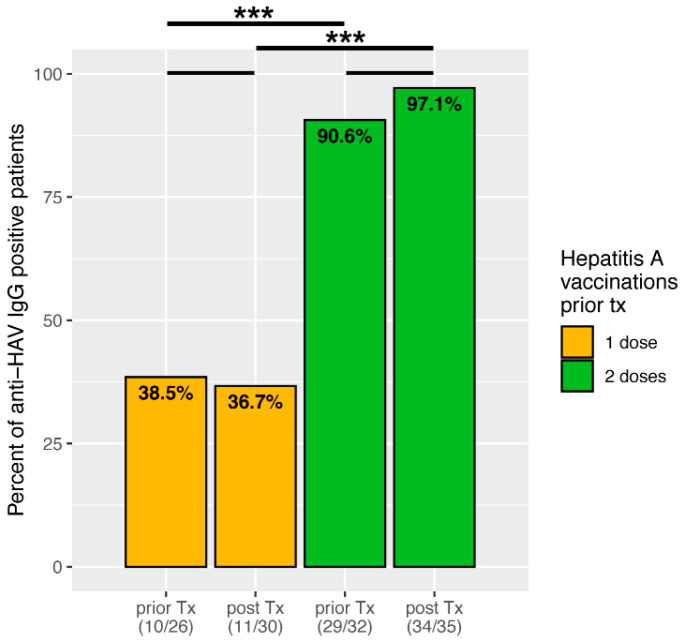
Bar plots showing percentage of anti-HAV-IgG positive patients, depending on number of vaccine doses prior to transplantation. Significant data is marked as follows: *** *p* < 0.001.

**Figure 2 children-09-01953-f002:**
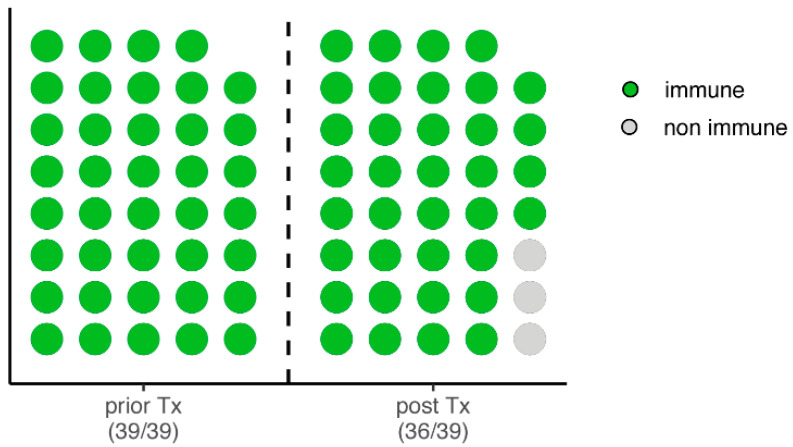
Dot plot of the 39 children who are anti-HAV-IgG positive (immune) before liver transplantation with their intraindividual course at the first annual follow-up after transplantation.

**Figure 3 children-09-01953-f003:**
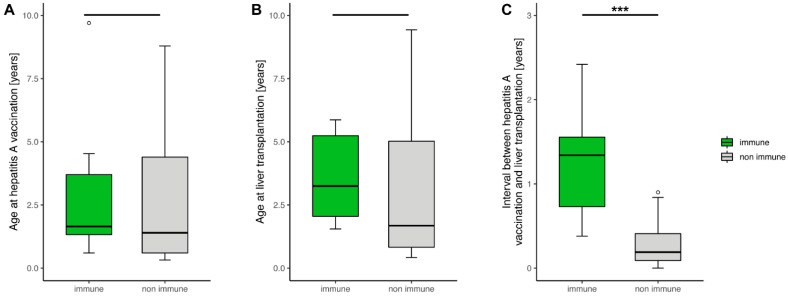
In the boxplots, patient data with only one hepatitis A vaccine dose prior to liver transplantation was examined. Patients were classified into immune and non immune on the basis of anti-HAV-IgG positivity at the first annual check-up after transplantation. The median and interquartile range in years of (**A**) age at vaccination, (**B**) age at liver transplantation and (**C**) intervals from vaccination to liver transplantation are shown. Only (**C**) shows a significant difference (*p* = 0.00008). Significant data is marked as follows: *** *p* < 0.001.

**Table 1 children-09-01953-t001:** Patient data on 65 children who received liver transplant between January 2003 and June 2021 at Hannover Medical School (Germany) and were administered at least one prior hepatitis A immunisation according to vaccination record.

	All Children Included (*n* = 65)
Gender, male (%)/female (%)	23 (35.4%)/42 (64.6%)
Diagnosis	Biliary atresia: 30 (46.2%) Progressive familial intrahepatic cholestasis: 13 (20%) Cystic fibrosis: 5 (7.7%) Cryptogenic cirrhosis: 5 (7.7%) Alagille syndrome: 4 (6.2%) Metabolic conditions: 3 (4.6%) Hepatic malignancy: 1 (1.5%) Acute liver failure: 1 (1.5%) Primary sclerosing cholangitis: 1 (1.5%) KIF12 variant: 1 (1.5%) Alpha-1 antitrypsin deficiency: 1 (1.5%)
Number of hepatitis A doses prior to transplantation	1 dose: 30 (46.1%) 2 doses: 35 (53.8%)
Age at 1st hepatitis A dose prior to transplantation in years, median (IQR)	2.1 (1.2–4.1)
Age at 2nd hepatitis A dose prior to transplantation in years, median (IQR)	3.1 (2.1–5.1)
Age at time of transplant in years, median (IQR)	4.6 (2.0–7.8)

**Table 2 children-09-01953-t002:** Overview of vaccinated and anti-HAV-IgG positive children before liver transplantation. The classification into immune and non-immune is based on the anti-HAV-IgG positivity at the first year of follow-up to investigate factors influencing the loss of immunity.

	Immune (*n* = 36)	Non-Immune (*n* = 3)	*p*
Gender, male (%)/female (%)	13 (36.1%)/23 (63.9%)	0 (0.0%)/3 (100.0%)	0.5377
Completed basic immunisation with 2 vaccine doses	29 (80.6%)	0 (0.0%)	<0.0131
Occurrence of biopsy-proven acute rejection (%)	14 (38.9%)	3 (100.0%)	0.0744
Vaccination administered below 12 months of age (%)	1 (2.8%)	1 (33.3%)	0.1498
Immunosuppression			
Tacrolimus (%)	29 (80.6%)	2 (66.7%)	0.5082
Ciclosporine (%)	7 (19.4%)	1 (33.3%)	0.5082
Intensified immunosuppression, calcineurin inhibitor + prednisolone or mycophenolate mofetil	15 (41.7%)	3 (100.0%)	0.0893

**Table 3 children-09-01953-t003:** Overview of children vaccinated before liver transplantation classified into immune and non-immune on the basis of anti-HAV-IgG positivity at the first annual check-up after transplantation.

	Immune (*n* = 45)	Non-Immune (*n* = 20)	*p*
Gender, male (%)/female (%)	16 (35.6%)/29 (64.4%)	7 (35.0%)/13 (65.0%)	0.966
Completed basic immunisation with 2 vaccine doses	34 (75.6%)	1 (5.0%)	<0.00001
Age at 1st hepatitis A dose, median (IQR)	2.14 (1.35–4.09)	1.48 (0.60–4.40)	0.215
Age at liver transplantation, median (IQR)	5.54 (3.13–8.19)	1.84 (0.81–5.05)	0.0008
Interval between last hepatitis A dose and liver transplantation, median (IQR)	1.41 (0.56–2.65)	0.18 (0.07–0.41)	<0.00001
Occurrence of biopsy-proven acute rejection (%)	15 (50.0%)	8 (40.0%)	0.648
Vaccination administered below 12 months of age (%)	1 (2.2%)	6 (30.0%)	0.000855
Immunosuppression			
Tacrolimus (%)	33 (73.3%)	12 (60.0%)	0.282
Tacrolimus trough level, median (IQR) [µg/L]	5.0 (3.9–6.5)	4.4 (3.8–4.7)	0.150
Ciclosporine (%)	12 (26.7%)	8 (40.0%)	0.282
Ciclosporine trough level, median (IQR) [µg/L]	131 (110–171)	118 (113–163)	0.881
Intensified immunosuppression, calcineurin inhibitor + prednisolone or mycophenolate mofetil	19 (42.2%)	12 (60.0%)	0.185

## Data Availability

All data requests should be submitted to the corresponding author for consideration. Access to anonymized data may be granted, following review.

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
