# Peer review of "Hepatitis A Immunity and Paediatric Liver Transplantation—A Single-Centre Analysis"

_children, 2022, doi:10.3390/children9121953_

Round 1
Reviewer 1 Report
The hepatitis A vaccine has been studied extensively and is safe, as the protective antibodies are demonstrated over 20 years after vaccination. Patients with chronic liver disease are considered to be at risk for severe hepatitis A virus infection. Children with liver disease undergoing transplantation are characterized by an impaired immune response, resulting in the loss of post-vaccination antibodies. In the submitted manuscript “Hepatitis A immunity and paediatric liver transplantation - A single-centre analysis”, the authors evaluated the presence of anti-HAV IgG antibodies (as a marker for vaccine immunogenicity) in a group of 65 liver transplant pediatric recipients. The authors analyzed the presence of protective antibodies detected on the admission of the children for liver transplantation and one year after the transplantation. The study on this topic is of great interest to the field, nevertheless, the present manuscript needs some improvements to be suitable for publication in "Children". The Results are difficult to follow and details are sometimes missing to understand the obtained data. The authors discuss "anti-HAV-IgG levels", but nowhere in the text are their values mentioned. I recommend that the authors cite measured anti-HAV IgG values or discuss their prevalence. Also, it would be of great benefit to the manuscript if authors could track the change in antibody values or their disappearance when comparing immune children at baseline – before transplantation and post-transplantation, with the following immunosuppression. It is confusing during the reading that the authors use the phrase "one vaccine" instead of "one dose", which I recommend clarifying. My specific recommendations to the authors are as follows.
For the Abstract
Lines 13-15: The sentence is difficult to understand; I recommend the authors paraphrase it.
Line 19: Most likely the authors mean that the children were vaccinated with one dose, so I recommend using "vaccinated with one dose" or a similar expression. This applies to the entire text of the manuscript.
For the Introduction section
Line 49-50: I recommend the authors unify the presentation of data as a number of children or percentage.
Lines 78-79: The sentence is hard to understand
For the Materials and Methods section
Lines 78-79: The sentence is difficult to understand, I recommend the authors edit it and include it in the paragraph with exclusion criteria.
For Results section
Lines 97-98: I recommend that the authors be more precise in their statements. It is not clear whether the children were vaccinated against HAV once with the full schedule or had only one dose of the vaccine administered.
Line 101: As the number of children (from the target population) is already specified in the previous paragraph, I recommend the authors delete "Of these 65 children…" from the sentence. It would also be more appropriate to include the sentence in the previous paragraph.
Line 106: I recommend the authors move the sentence" Median age at transplantation was 4.6 years (2.0-7.8)" in the previous paragraph after the gender distribution.
Table 1: I recommend including distribution by the male gender, also.
"1 vaccination/2 vaccinations" is to be changed to "1 dose/ 2 doses"
Line 113: I recommend citing Table 1 at the end of the sentence (which allows the reader to quickly navigate the number of children in the two subgroups).
Lines 114-115: I recommend the authors give the values of anti-HAV IgG titers especially those with a protective role (sufficient protection).
Line 116: I recommend „had a titre measure“ be changed to “had ant-HAV IgG” or “were anti-HAV IgG positive”
Line 117: I recommend editing the sentence and using the phrase "compared to" (one dose vs two doses)
Figure 1: Lines 127-130 are already described in the main text
Line 135: I recommend the authors change "...anti-HAV-IgG levels" to "...anti-HAV-IgG positivity" keeping the meaning of the sentence
Line 136: I recommend the authors change „…number of vaccinations“ to „…number of doses“ keeping the meaning of the sentence
Table 2: I recommend including distribution by the male gender, also.
I recommend where necessary replacing "vaccination" with "dosses"
Lines 140-141: I recommend the authors argue (be more clear) in their statement "were less likely to have been vaccinated against hepatitis A for the first time in their first year of life" since the HAV vaccine is for children older than one year, which they specify several times in the text. In this connection, I recommend citing in the Discussion recommendations of guidelines for earlier vaccination against HAV of children undergoing liver transplantation or immunosuppression.
Lines 142-143: It is difficult to understand the meaning of the sentence in the context of the manuscript.
Lines 154-157: The sentence is hard to understand.
Figure 2/line 160: The first group of children is "immune" and the second is "non-immune".
For Discussion
Lines 175-176: I recommend that authors use either numbers only (three quarters) or percentages only (94%) when describing data, otherwise it is difficult to follow the relationship.
Lines 179, 189, 194, 212: In this case, the expression "anti-HAV-IgG levels" cannot be used because the authors nowhere mention antibody titer values. I recommend using "Anti-HAV IgG positive"
Line 204: I recommend that the authors revise and edit the sentence "However, as up to 30% of children receive liver transplants in their first year of life...." as it is understood that 30% of all children, in general, receive a liver transplant
Author Response
Please see the attachment.
Dear Editors,
First of all, we would like to thank the reviewers for their detailed and helpful
comments on “Hepatitis A immunity and paediatric liver transplantation — A
single-centre analysis”. We would like to submit a revised version of the manuscript that takes into account the comments made by the referees.
Please find the detailed answers to the reviewers' comments inserted just below the respective remarks.
Point 1: “The authors discuss "anti-HAV-IgG levels", but nowhere in the text are their values mentioned. I recommend that the authors cite measured anti-HAV IgG values or discuss their prevalence.”
Response: Thank you for this important comment. We have added paragraph 2.2 to the Materials and Methods section. Here we describe the chemiluminescence
microparticle immunoassay used by our laboratory for the determination of hepatitis A immunity. This method uses a chemiluminescent signal to detect anti-HAV-IgG and compares it to a cut-off signal from an active calibration. According to the manufacturer’s instructions, a ratio of ≥1.0 indicates a positive result and thus the detection of anti-HAV-IgG resulting in immunity.
Unfortunately, only "positive" and "negative" results are recorded by the laboratory, so that no titres/ratios are available and no detailed statements can be made about this. However, this is an important point that should be considered in future studies, as studies have shown poorer titer response in children with liver disease [1]. We also extended the discussion to better clarify this point.
Point 2: “Also, it would be of great benefit to the manuscript if authors could track the change in antibody values or their disappearance when comparing immune children at baseline – before transplantation and post-transplantation, with the following immunosuppression.”
Response: Thank you for this helpful comment. We have added paragraph 3.3
(Intraindividual change in hepatitis A immunity from prior to post liver transplantation) to the results section. Here we have examined the children who were anti-HAV-IgG positive at the time of liver transplantation and can show that around 92.3% are still protected at the first annual check-up. Unfortunately, the number of non-immune children is very small, so that only Fisher's exact test could be used and no Mann–Whitney U. Again, a single vaccination before transplantation is a significant risk factor for non-immunity.
Point 3: “It is confusing during the reading that the authors use the phrase "one
vaccine" instead of "one dose", which I recommend clarifying.”
Response: Thank you for this correction. We changed it accordingly.
Point 4: “Lines 13-15: The sentence is difficult to understand; I recommend the
authors paraphrase it.”
Response: We paraphrased it and hopefully it is now easier to understand.
Point 5: “Line 49-50: I recommend the authors unify the presentation of data as a number of children or percentage.”
Response: Thank you for this helpful comment. We unified the data presentation
accordingly.
Point 6: “Lines 78-79: The sentence is hard to understand”
Response: Thank you for this important comment. We have added paragraph 2.2 to the Materials and Methods section to describe in more detail the chemiluminescence microparticle immunoassay used by our laboratory for the determination of hepatitis A immunity.
Point 7: “Lines 78-79: The sentence is difficult to understand, I recommend the
authors edit it and include it in the paragraph with exclusion criteria.”
Response: We changed it accordingly and added it to the exclusion criteria.
Point 8: “Lines 97-98: I recommend that the authors be more precise in their
statements. It is not clear whether the children were vaccinated against HAV once with the full schedule or had only one dose of the vaccine administered.”
Response: Thank you for the correction. We clarified the point in the text as follows: “A total of 58 of 139 (41.7%) eligible children were vaccinated with at least one dose against hepatitis A before liver transplantation”
Point 9: “Line 101: As the number of children (from the target population) is already specified in the previous paragraph, I recommend the authors delete "Of these 65 children…" from the sentence. It would also be more appropriate to include the sentence in the previous paragraph.”
Response: Agreed, we changed it accordingly.
Point 10: “Line 106: I recommend the authors move the sentence" Median age at
transplantation was 4.6 years (2.0-7.8)" in the previous paragraph after the gender distribution.”
Response: Agreed, we changed it accordingly.
Point 11: “Table 1: I recommend including distribution by the male gender, also. "1 vaccination/2 vaccinations" is to be changed to "1 dose/ 2 doses"”
Response: Thank you for the correction. We added the male gender changed the
expression of vaccination in dose(s).
Point 12: “Lines 114-115: I recommend the authors give the values of anti-HAV IgG titers especially those with a protective role (sufficient protection).”
Response: Thank you for this important comment. Unfortunately, due to the
chemiluminescence microparticle immunoassay used, only anti-HAV-IgG "positive" and "negative" results are available. Therefore, an analysis of titres cannot be carried out.
Point 13: “Line 116: I recommend „had a titre measure“ be changed to “had ant-HAV IgG” or “were anti-HAV IgG positive””
Response: Agreed, we changed it accordingly.
Point 14: “Line 117: I recommend editing the sentence and using the phrase
"compared to" (one dose vs two doses)”
Response: Thank you for the correction, we changed it accordingly.
Point 15: “Figure 1: Lines 127-130 are already described in the main text”
Response: We removed it to make it easier to follow the description.
Point 16: “Line 135: I recommend the authors change "...anti-HAV-IgG levels" to
"...anti-HAV-IgG positivity" keeping the meaning of the sentence”
Response: Thank you for the correction. We adapted it accordingly.
Point 17: “Line 136: I recommend the authors change „…number of vaccinations“ to„…number of doses“ keeping the meaning of the sentence”
Response: Thank you for the correction. We adapted it accordingly.
Point 18: “Table 2: I recommend including distribution by the male gender, also.”
Response: Thank you for the comment. We have added the male gender to the table.
Point 19: “Lines 140-141: I recommend the authors argue (be more clear) in their
statement "were less likely to have been vaccinated against hepatitis A for the first time in their first year of life" since the HAV vaccine is for children older than one year, which they specify several times in the text. In this connection, I recommend citing in the Discussion recommendations of guidelines for earlier vaccination against HAV of children undergoing liver transplantation or immunosuppression.”
Response: Thank you for the helpful comment. We clarified the point in the text as follows: “Children who are not immune received vaccination more often below 12 months of age (p=0.000855).” Moreover, we added a paragraph about the CDC and ISDA recommendations for pediatric patients in the Discussion.
Point 20: “Lines 142-143: It is difficult to understand the meaning of the sentence in the context of the manuscript.”
Response: Thank you for the comment. We have worded the sentence more clearly:“There were no differences in gender and trough levels of immunosuppression(including an intensified immunosuppression). Moreover, history of biopsy-proven acute rejection did not influence HAV-immunity.”
Point 21: “Lines 154-157: The sentence is hard to understand.”
Response: Thank you for this comment. We tried to clarify it and changed the text as follows: ”A significant influence of immunosuppression, history of biopsy-proven acute rejection and gender on non-immunity could not be demonstrated. Moreover,both groups did not differ in the number of children vaccinated under 12 months of age.”
Point 22: “Figure 2/line 160: The first group of children is "immune" and the second is "non-immune".”
Response: Thank you for the correction. We changed it accordingly.
Point 23: “Lines 175-176: I recommend that authors use either numbers only (three quarters) or percentages only (94%) when describing data, otherwise it is difficult to follow the relationship.”
Response: Thank you for the correction. We adapted it accordingly.
Point 24: “Lines 179, 189, 194, 212: In this case, the expression "anti-HAV-IgG
levels" cannot be used because the authors nowhere mention antibody titer values. I recommend using "Anti-HAV IgG positive"”
Response: Thank you for the important correction. We changed it accordingly at all four points.
Point 25: “Line 204: I recommend that the authors revise and edit the sentence
"However, as up to 30% of children receive liver transplants in their first year of life...." as it is understood that 30% of all children, in general, receive a liver transplant”
Response: Thank you for the comment. We changed the text as follows: “As up to 30% of children with liver disease receive liver transplants in their first year of life”.
We hope that you will find the changes we made to the manuscript satisfactory, and that you will consider our manuscript for publication.
Yours sincerely
References:
1. Ferreira, C.T.; da Silveira, T.R.; Vieira, S.M.; Taniguchi, A.; Pereira-Lima, J. Immunogenicity and safety of hepatitis A vaccine in children with chronic liver disease. J Pediatr Gastroenterol Nutr 2003, 37, 258-261, doi:10.1097/00005176-200309000-00011.

Reviewer 2 Report
The manuscript entitled ¨Hepatitis A immunity and pediatric liver transplantation — _A single-center analysis¨ by Tobias Laue et al, aims at examining the immunization status against HAV in pediatric liver transplant candidates at the time of transplantation and one year later.
The manuscript is well organized, and the data provided here is of interest to the field of transplantation.
There are however some issues that need to be addressed.
1. What are the criteria utilized to determine which patients were immune and non-immune?
2. What is the meaning of sufficient protection? this term is very vague; it needs to be supported by the data corresponding to the antibody titers. Therefore, please provide the different antibody titters obtained for each patient and clarify how the authors determined what was sufficient protection vs non-sufficient. Was there a cut-off value for this determination?
3. Based on the added data regarding points 1 and 2, include a discussion point about the data.
4. There are several grammatical errors and typos in the manuscript that need to be corrected.
Author Response
Please see the attachment.
Dear Editors,
First of all, we would like to thank the reviewers for their detailed and helpful
comments on “Hepatitis A immunity and paediatric liver transplantation — A
single-centre analysis”. We would like to submit a revised version of the manuscript that takes into account the comments made by the referees.
Please find the detailed answers to the reviewers' comments inserted just below the respective remarks.
Points 1 - 3: “What are the criteria utilized to determine which patients were immune and non-immune? What is the meaning of sufficient protection? this term is very vague; it needs to be supported by the data corresponding to the antibody titers. Therefore, please provide the different antibody titters obtained for each patient and clarify how the authors determined what was sufficient protection vs non-sufficient. Was there a cut-off value for this determination? Based on the added data regarding points 1 and 2, include a discussion point about the data.”
Response: We have added paragraph 2.2 to the Materials and Methods section.
Here we describe the chemiluminescence microparticle immunoassay used by our laboratory for the determination of hepatitis A immunity. This method uses a chemiluminescent signal to detect anti-HAV-IgG and compares it to a cut-off signal from an active calibration. According to the manufacturer’s instructions, a ratio of ≥1.0 indicates the detection of anti-HAV-IgG resulting in immunity.
Unfortunately, only "positive" and "negative" results are recorded by the laboratory, so that no titres are available and no detailed statements can be made about this. However, this is an important point that should be considered in future studies, as studies have shown poorer titer response in children with liver disease [1]. We also extended the discussion to better clarify this point.
Point 4: “There are several grammatical errors and typos in the manuscript that need to be corrected.”
Response: We apologise for these mistakes. The manuscript has been proofread
again and is now hopefully easier to understand. We hope that you will find the changes we made to the manuscript satisfactory, and that you will consider our manuscript for publication.
Yours sincerely
References:
1. Ferreira, C.T.; da Silveira, T.R.; Vieira, S.M.; Taniguchi, A.; Pereira-Lima, J. Immunogenicity and safety of hepatitis A vaccine in children with chronic liver disease. J Pediatr Gastroenterol Nutr 2003, 37, 258-261, doi:10.1097/00005176-200309000-00011.

Round 2
Reviewer 1 Report
I have read the edited manuscript “Hepatitis A immunity and paediatric liver transplantation - A single-centre analysis”, provided to me. The authors have taken into account the comments and recommendations made, which has led to a significant improvement of the manuscript. After my second review, I have a few more minor comments. The line numbering used is from the cleaned version.
Line 13: hepatitis A virus (HAV) is often asymptomatic and limited in children. it is much more correct to say: self-limited
Lines 15-16: “At the same time” instead of “In contrast”
line 22: After one or two doses, respectively 38.5% and 90.6% of 65………
line 23: “This remained stable up..”, I recommend the authors include “percent” - “This percent remains stable up..”
line 34: I recommend to include “asymptomatic course of infection…”
line 45: I recommend to delete “in tern”
line 85: I recommend to delete “list number 6C29”
line 87: I recommend the authors replace ‘therefore” with “Briefly/ In brief,…”
lines 93-94: the sentence “Only whether the S/CO is positive or negative is recorded by the laboratory system, but not the S/CO”. I recommend the authors replace it with “The results are presented qualitatively - positive or negative.”
Line 111: I recommend to delete “However” from the beginning of the sentence
Lines 114-118: I recommend that the authors merge the two sentences, delete “Further diagnoses in order of frequency were”, and correct order “20% - with progressive familial intrahepatic cholestasis (PFIC), 7.7% - with cystic fibrosis ....”
Lines 129-130: I recommend that the authors merge the two sentences, and instead of "Here" use ", of whom".
Lines 135-136: the sentence is not properly worded. I recommend the following option: “However, the number/percent (please, choose one of the options) of anti-HAV protected patients is significantly smaller compared to those ......”
Figure 1: I recommend the axis title: “percent of anti-HAV IgG positive patients” and in the figure caption I recommend the authors replace ‘numbers” with “percent”
Line 153: recommend adding at the end of the sentence “the transplantation”
Author Response
Thank you for the valuable comments!
We have revised the manuscript according to the comments.
We hope that you will find the changes we made to the manuscript satisfactory, and that you will consider our manuscript for publication.
Reviewer 2 Report
The authors have successfully addressed the comments and the manuscript has been significantly improved. Despite the low sample size, the study provides important information.
Author Response
Thank you for the valuable comments!